# Performance of Scoring Systems in Predicting Clinical Outcomes of Patients with Emphysematous Pyelonephritis: A 14-Year Hospital-Based Study

**DOI:** 10.3390/jcm11247299

**Published:** 2022-12-08

**Authors:** Chun-Cheng Chen, Ming-Shun Hsieh, Sung-Yuan Hu, Shih-Che Huang, Che-An Tsai, Yi-Chun Tsai

**Affiliations:** 1Department of Emergency Medicine, Taichung Veterans General Hospital, Taichung 40705, Taiwan; 2Department of Emergency Medicine, Taipei Veterans General Hospital, Taoyuan Branch, Taoyuan 330, Taiwan; 3Department of Emergency Medicine, Taipei Veterans General Hospital, Taipei 11217, Taiwan; 4School of Medicine, National Yang Ming Chiao Tung University, Taipei 11221, Taiwan; 5Department of Post-Baccalaureate Medicine, College of Medicine, National Chung Hsing University, Taichung 402, Taiwan; 6School of Medicine, Chung Shan Medical University, Taichung 40201, Taiwan; 7Institute of Medicine, Chung Shan Medical University, Taichung 40201, Taiwan; 8Department of Emergency Medicine, Chung Shan Medical University Hospital, Taichung 40201, Taiwan; 9Lung Cancer Diagnosis and Treatment Research Center, Chung Shan Medical University Hospital, Taichung 40201, Taiwan; 10Division of Infectious Disease, Department of Internal Medicine, Taichung Veterans General Hospital, Taichung 40705, Taiwan

**Keywords:** emphysematous pyelonephritis (EPN), Mortality in Emergency Department Sepsis (MEDS) score, National Early Warning Score (NEWS), Rapid Emergency Medicine Score (REMS), scoring systems, receiver operating characteristic curve (ROC)

## Abstract

Background: Emphysematous pyelonephritis (EPN) is a rare but severe necrotizing infection causing there to be gas in the pelvicalyceal system, renal parenchyma, and perirenal or pararenal space. Physicians should attend to EPN because of its life-threatening septic complications. The overall mortality rate has been reported to be as high as 20–40%. In addition, most patients had diabetes mellitus (DM) and obstructive uropathy. The most common isolated microorganism is *Escherichia coli*. This study aims to analyze the risk factors and performance of scoring systems in predicting the clinical outcomes of patients with EPN. Materials and Methods: We collected the data of patients with EPN in this single hospital-based retrospective study from the electronic medical records of Taichung Veterans General Hospital between January 2007 and December 2020. Radiological investigations of abdominal computed tomography (CT) confirmed the diagnosis of EPN. In addition, we analyzed demographics, clinical characteristics, and laboratory data. Finally, we used various scoring systems to predict clinical outcomes. Results: A total of fifty patients with EPN, whose diagnoses were confirmed through CT, were enrolled in the study. There were 18 males (36%) and 32 females (64%), with a mean age of 64.3 ± 11.3 years. The in-hospital mortality rate was 16%. A DM of 34 (68%) patients was the most common comorbidity. Fever was the most common symptom, found in 25 (50%) patients. The Mortality in Emergency Department Sepsis (MEDS) score was 4.64 ± 3.67 for survivors and 14.25 ± 5.34 for non-survivors (*p* < 0.001). The National Early Warning Score (NEWS) was 3.64 ± 2.33 for survivors and 7.13 ± 4.85 for non-survivors (*p* = 0.046). The Rapid Emergency Medicine Score (REMS) was 5.81 ± 1.97 for survivors and 9.13 ± 3.87 for non-survivors (*p* = 0.024). Regarding performance of mortality risk prediction, the AUC of ROC was 0.932 for MEDS, 0.747 for REMS, and 0.72 for NEWS. Conclusions: MEDS, REMS, and NEWS could be prognostic tools for the prediction of the clinical outcomes of patients with EPN. MEDS showed the best sound performance. In those with higher scores in MEDS (≥12), REMS (≥10), and NEWS (≥8), we recommended aggressive management and appropriate antimicrobial therapy as soon as possible to reduce mortality. Further large-scale studies are required to gain a deep understanding of this disease and to ensure patient safety.

## 1. Introduction

Emphysematous pyelonephritis (EPN) is a severe and acute necrotizing infection of the kidney, leading to an accumulation of gas in the pelvicalyceal system, renal parenchyma, and perirenal or pararenal space. The earliest case of the accumulationof gas due to renal infection was reported in 1898 by Kelly and MacCallum [1,2]. Schultz and Klorfein coined the term EPN and described the severe illness in 1962 [3]. It was a rare illness that resulted in death due to septic complications. The current meta-analysis reported mortality rates of up to 20–40% [4]. Radiological investigation methods, including abdominal ultrasound (US) and computed tomography (CT), can confirmthe diagnosis of EPN [5].

Enteric gram-negative facultative anaerobes arecommon microorganisms that can cause EPN, by initiating thefermentation of glucose within the urine to produce gases, including nitrogen, hydrogen (H_2_), and carbon dioxide (CO_2_), resulting in emphysematous necrosis. Most patients had diabetes mellitus (DM), urolithiasis, urinary tract obstructions, or chronic kidney disease. The high levels of tissue glucose and decreased tissue perfusion caused by the aforementioned illnesses provided a suitable environment for bacteria growth and the production of CO_2_ and H_2_ [6,7]. Delayed treatment can result in a high mortality rate. The conventional methods of treatment used in EPN patients, including nephrectomy and open drainage, had a historically high mortality of up to 78% [8]. Some patients who received conventional medicine could not preserve renal function and needed hemodialysis. The minimally invasive procedures, such as percutaneous drainage (PCD), were introduced for patients with EPN by Hudson et al. and have, significantly, led to a substantially lower mortality rate since 1986 [9]. Currently, most experts advocate for aggressive medical treatment and minimally invasive procedures, such as PCD or the implantation of a double-J stent, followed by delayed nephrectomy as the standard method for managing EPN [7].

The Glasgow Admission Prediction Score (GAPS), Mortality in Emergency Department Sepsis (MEDS) score, Modified Early Warning Score (MEWS), National Early Warning Score (NEWS), Rapid Acute Physiology Score (RAPS), Rapid Emergency Medicine Score (REMS), quick Sepsis Related Organ Failure Assessment (qSOFA), and Worthing Physiological Scoring system (WPS) are the different scoring systems, consisting of various available clinical parameters, used to predict the clinical outcomes of patients in emergency and critical care [10,11,12,13,14,15]. Due to the low incidence but high mortality rate of EPN, we aimed to validate the performance of various scoring systems (*n* = 8) in assessing the severity and clinical outcomes of EPN. Therefore, we investigated the clinical characteristics of EPN and applied multiple scoring systems to predict the risk of mortality.

## 2. Materials and Methods

### 2.1. Data Collection and Definition

The institutional review board approved this study concerning Taichung Veterans General Hospital (TCVGH) (No. CE21215A), Taichung, Taiwan. It was a single hospital-based retrospective study of patients with EPN confirmed by CT in the emergency department (ED). Patients’ data, including demographics, laboratory investigations, and clinical outcomes, were extracted from the electronic medical records of TCVGH from January 2007 to December 2020. The primary outcome was in-hospital mortality. We used univariate and multivariate analyses to evaluate the risk of mortality.

### 2.2. Scoring Systems

We used eight scoring systems, including GAPS, MEDS score, MEWS, NEWS, RAPS, REMS, qSOFA, and WPS, to analyze the clinical outcomes and predict the mortality risk.

### 2.3. Statistical Analysis

Continuous data were expressed as mean ± standard deviation (SD). Categorical data were expressed as numbers and percentages. Chi-square tests were used to compare absolute features, and Mann–Whitney–Wilcoxon U-tests were used to compare continuous features regarding their effects on mortality risks in the survivors and non-survivors. We conducted univariate and multivariate analyses using the Cox regression model to assess possible predictors for mortality and results were expressed as hazard ratio (HR) and confidence interval (CI). We used the area under the curve (AUC) and receiver operating characteristic curve (ROC) to compare predictive powers across the various scoring systems. We used cut-off points to stratify the mortality risk in terms of sensitivity, specificity, negative predictive value (NPV), and positive predictive value (PPV). The *p* values < 0.05 were considered statistically significant. We performed analyses using the Statistical Package for the Social Science (IBM SPSS version 22.0; International Business Machines Corp., New York, NY, USA).

### 2.4. Image Classification

In this study, we classified EPN as follows: (1) Class 1: gas in the collecting system only; (2) Class 2: gas in the renal parenchyma without extension to extrarenal space; (3) Class 3A: extension of gas or abscess to perinephric space; Class 3B: extension of gas or abscess to pararenal space; and (4) Class 4: bilateral EPN or solitary kidney with EPN according to the classification of EPN by the study of Huang and Tseng in 2000 [16].

## 3. Results

### 3.1. Demographics and Clinical Characteristics

We summarized the demographics and clinical findings of 50 patients with EPN in Table 1. The mean age was 64.33 ± 11.27 years with a male-to-female ratio of 1:1.8 (18:32). The mean length of hospital stay was 25.58 ± 16.82 days. Most patients had severe infection conditions in the ED, including 25 (50%) patients with systemic inflammatory response syndrome (SIRS) or shock, 17 (34%) patients with respiratory failure, and admission to the intensive care unit (ICU). Age (62.52 ± 10.91 vs. 73.82 ± 8.34, *p* = 0.008) and gastrointestinal (GI) disease (28.6% vs. 75%, *p* = 0.019) were significantly higher in the non-survivors. Eight (16%) patients died during hospitalization. The most common comorbidities were DM in 34 (68%) patients, followed by cardiovascular disease in 30 (60%) patients, chronic renal failure in 20 (40%) patients, GI disease in 18 (36%) patients, and genitourinary (GU) diseases in 15 (30%) patients. Although the most common risk factors of EPN were DM (68%) and urinary tract obstruction (32%), there were no statistical differences between the survivors and non-survivors with DM and obstructive uropathy.

### 3.2. Clinical Syndromes and Management

Fever was the most common symptom (50%), followed by flank pain (40%), and GI symptoms (30%). Symptoms, including fever, flank pain, and conscious change, showed no significant differences between the survivor and non-survivor groups (Table 1).

### 3.3. Laboratory Data and Scoring Systems

Laboratory data and scoring systems were summarized in Table 2 and Table 3. Blood urea nitrogen (BUN) (39.00 ± 36.37 vs. 51.50 ± 18.42, *p* = 0.027) and lactate (21.26 ± 20.78 vs. 50.44 ± 33.32, *p* = 0.02) were significantly higher in the non-survivors. The scores of MEDS (4.64 ± 3.67 vs. 14.25 ± 5.34, *p* < 0.001), NEWS (3.64 ± 2.33 vs. 7.13 ± 4.85, *p* = 0.046), and REMS (5.81 ± 1.97 vs. 9.13 ± 387, *p* = 0.024) were significantly higher in the non-survivors, unlike GAPS (20.74 ± 2.63 vs. 18.00 ± 3.55, *p* = 0.023). In multivariate logistic regression analyses, SpO_2_, age, lactate dehydrogenase (LDH), and lactate significantly differed between the survival and non-survival patients. The scoring systems of GAPS, MEDS, NEWS, and WPS varied considerably.

### 3.4. Microbiology

The positive rate of blood culture was 27 (60.0%) of 45 patients. The positive rate of urine culture was 35 (72.9%) of 48 patients. The positive rate of pus or tissue was 29 (90.6%) of 32 patients. There were no significant differences between the survivors and non-survivors in the rates of the blood, urine, pus, or tissue cultures. The leading microorganisms of cultures from urine (*n* = 20), blood (*n* = 13), and pus/tissue (*n* = 14) were *Escherichia coli*. Other microorganisms included *Klebsiella pneumoniae*, *Pseudomonas aeruginosa*, *Staphylococcus aureus*, *Enterobacter cloacae*, *Streptococcus pneumoniae*, *Enterococcus faecalis*, and *Candida albicans*. The results were summarized in Table 4.

### 3.5. Clinical Management and Course

The clinical management of EPN included conservative treatment (antibiotics only), percutaneous drainage (PCD), and surgical debridement. All of those revealed no significant difference between the survivor and non-survivor groups. Incidence of shock, respiratory failure, and ICU admission was higher in the non-survivors than in the survivors. In addition, the antibiotics, management, and clinical course of EPN revealed no significant difference between the survivor and non-survivor groups (Table 5).

### 3.6. Univariate and Multivariate Analyses to Evaluate the Mortality Risk

We showed the results of univariate analyses for the predisposing factors of the clinical outcomes of EPN summarized in Table 6. We found the following: there were higher HRs of age, respiratory rate (RR), Glasgow coma scale (GCS), SpO_2_, lactate dehydrogenase (LDH), lactate, and prothrombin time (PT) in non-survivors than those in survivors. In addition, scores of MEDS, NEWS, RAPS, REMS, qSOFA, and WPS were significantly higher in the non-survivors than in survivors, with GAPS being the exception. The results of the multivariate Cox regression analyses for the clinical outcomes of EPN were summarized in Table 7. The scores of MEDS (*p* = 0.004), NEWS (*p* = 0.001), qSOFA (*p* = 0.043), and WPS (*p* = 0.001) had significantly higher HRs in non-survivors than in survivors, unlike GAPS (*p* = 0.0017).

### 3.7. Receiver Operating Characteristic Curve (ROC) of Scoring Systems

The AUROC of MEDS, REMS, GAPS, and NEWS were 0.932, 0.747, 0.747, and 0.720, respectively, to evaluate the accuracy of predicting the risks of mortality (Figure 1). The cut-off point of MEDS was 12, with an AUROC of 0.932. The cut-off point of REMS was 10, with an AUROC of 0.747. The cut-off point of GAPS was 22 with an AUROC of 0.747. The cut-off point of NEWS was 8 with an AUROC of 0.720. The AUC of ROC, cut-off point, sensitivity specificity, positive predictive value (PPV), negative predictive value (NPV), accuracy, and standard error of MEDS, REMS, NEWS, and GAPS to predict the risk of mortality were shown in Table 8.

### 3.8. Cumulative Survival Rates by Kaplan-Meier

We used the Kaplan–Meier analysis to calculate the cumulative survival rates of patients with EPN to predict the 30-day mortality risk. The cut-off points of MEDS (12), REMS (10), NEWS (8), and GAPS (22) demonstrated significant differences between survivors and non-survivors (Figure 2).

## 4. Discussion

EPN is an acute severe kidney infection that causes gas to form in the collection system, renal parenchyma, and perirenal or pararenal tissues [1,2]. EPN had a high mortality rate of up to 78% in the late 1970s. In recent decades, the introduction of minimally invasive procedures, such as PCD or percutaneous nephrostomy (PCN), has lowered the mortality rate to 21% [17,18]. DM is the most common underlying disease related to EPN [3,19]. Bacteremia and microorganisms from urine or pus are present in more than 50% of patients [16]. *Escherichia coli* is the most common Gram-negative facultative anaerobic microorganism in patients with EPN. Still, other microorganisms, such as *Klebsiella pneumoniae* and *Proteus mirabilis*, have also been reported [16,17,18]. Therefore, physicians should pay great attention to the clinical clues that point to EPN due to a higher mortality rate if there is a delay in diagnosis and aggressive management [19].

The various risk factors of EPN have been identified in the literature. The presence of serum creatinine levels of more than 1.4 mg/dl and a platelet count of less than 60,000/uL demonstrated a high predictive value for the mortality of EPN [20]. However, Michaeli et al. described that age, sex, site of infection, BUN, and blood glucose were not the prognostic factors of EPN [21]. Huang and Tseng et al. concluded that thrombocytopenia, acute kidney injury (AKI), disturbance of consciousness, and shock were associated with a high mortality rate or poor prognostic outcomes [16]. Falagas et al. reported that conservative medical treatment for bilateral EPN, type I EPN, and thrombocytopenia were associated with an increased mortality rate [19]. The association with thrombocytopenia is related to the development of disseminated intravascular coagulopathy and sepsis. Kapoor et al. associated altered mental status, thrombocytopenia, renal failure, and severe hyponatremia with an increased mortality rate [22]. In our study, there were no statistical significances in thrombocytopenia, AKI, and disturbance of consciousness. We found that age, GI disease, SIRS or shock, respiratory failure, admission to ICU, BUN, and lactate revealed significant differences between the survivors and non-survivors.

Numerous parameters, mandatory management, and different stages of the disease may impact the clinical outcomes. According to the contemporary meta-analysis, it is difficult to determine which treatment would be the best choice for EPN. The management of EPN consisted of three options: (1) medical management with antimicrobials alone; (2) surgical intervention (i.e., total or subtotal nephrectomy) with continued medical management; or (3) PCD with medical management and possible further surgical intervention if medical management fails [23]. The management decision-making for EPN depended on the response to antimicrobials, the preservation of renal function, and radiological findings. In our study, more than 50% of patients were treated with PCD with continued medical management and possible further surgical intervention. Some studies use CT images to guide the treatment of patients with EPN [16,17,18,19,20]. However, in our study, CT classification had no significant correlation with clinical outcomes. 

We used different scoring systems to analyze the risk factors and predict the clinical outcomes of patients with EPN. Not only the risk factors but also the scoring systems should be taken into consideration when deciding the most suitable treatment. In the literature review, Chawla et al. analyzed 90 consecutive patients and found that the NEWS scoring system was best able to predict the requirement of ICU care [24]. Krishnamoorthy et al. described that the objective and qualitative risk stratification assessment is necessary for treating patients with EPN [25]. 

The scoring systems of GAPS, MEDS, NEWS, qSOFA, and WPS have a significant capacity to predict mortality risk in patients with EPN. The best predictive tool is MEDS, with a high AUROC of 0.932 at the cut-off point of 12. MEDS is based on clinical history, features, laboratory data, and physiological measurements. The scoring systems can expedite the management of patients with risk factors complicating the severe illness. The Yap study described that the NEWS score is an ideal tool for an objective assessment, to evaluate the need for admission to ICU in patients with EPN in the ED [26]. The MEDS is excluded in Yap’s study because the variable of “terminal illness” has the highest weighting in MEDS. However, such patients are generally managed palliatively in the general ward rather than aggressively in the ICU.

## 5. Limitations

There were some limitations in our study. First, it was a single-center study of EPN. Second, it was a study of a small sample size and due to the long study duration, on average we enrolled less than five patients per year. Third, it was of a retrospective nature. Fourth, the infrequent nature of EPN limited our freedom to perform a prospective randomized control study. 

## 6. Conclusions

MEDS, REMS, and NEWS are promising scoring systems capable of predicting the clinical outcomes of patients with EPN. MEDS is an ideal tool for objective assessments, able to evaluate mortality risk in patients with EPN in the ED. It performs best in predicting the clinical outcomes of patients with ENP. Its simplicity and utility aid medical staff in the early recognition of critical patients and offer a time management guide to improve results. Further large-scale studies are required to gain a sound evidence base and ensure patient safety.

## Figures and Tables

**Figure 1 jcm-11-07299-f001:**
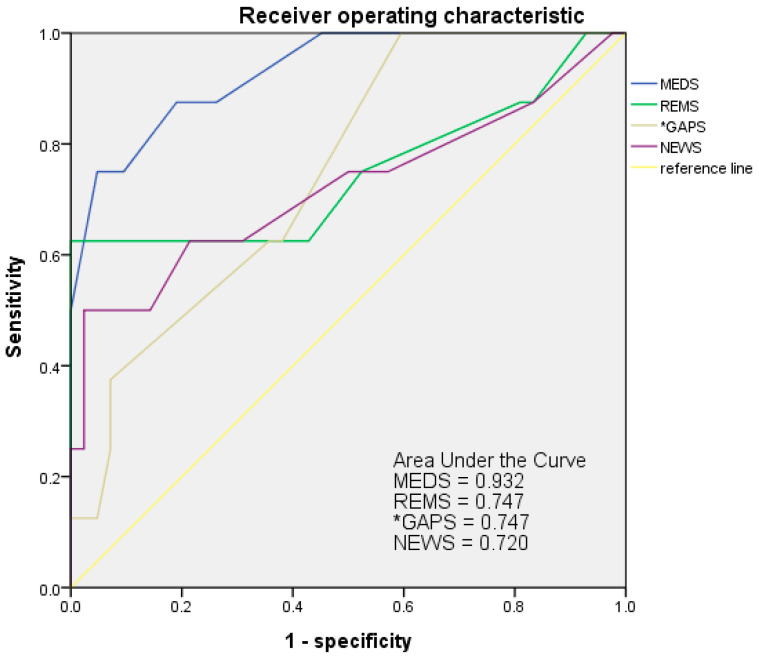
ROC of MEDS, REM, GAPS, and NEWS was analyzed to show the accuracy in predicting the mortality risks of patients with EPN. The AUC of ROC for MEDS, REMS, GAPS, and NEWS indicated 0.932, 0.747, 0.747, and 0.720, respectively. AUC = Area under the curve; ROC = Receiver operating characteristic curve.

**Figure 2 jcm-11-07299-f002:**
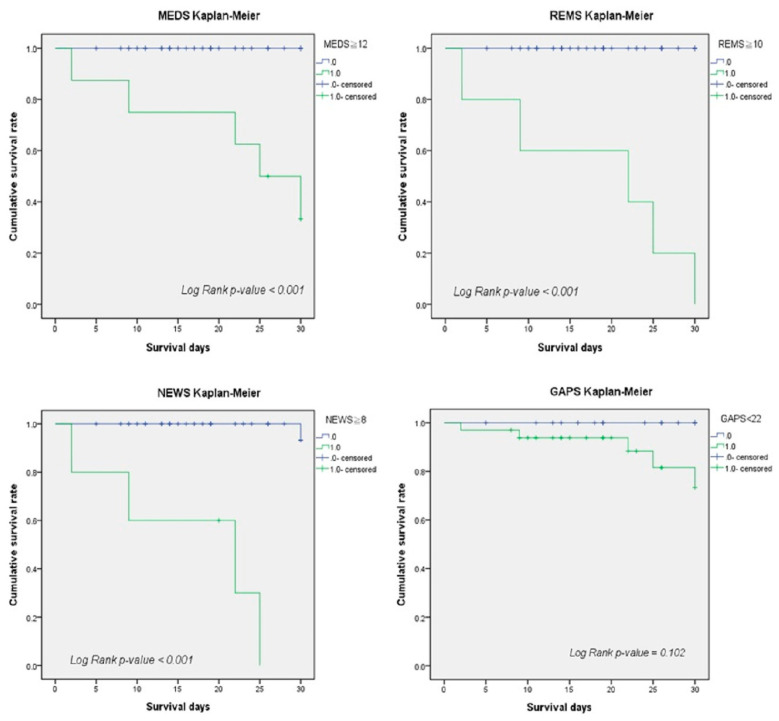
The cumulative survival rates of patients with EPN were calculated to predict the 30-day mortality rate by Kaplan–Meier analysis. The cut-off points of MEDS, REMS, NEWS, and GAPS were 12, 10, 8, and 22, respectively.

**Table 1 jcm-11-07299-t001:** Demographics and clinical characteristics of EPN.

General Data	All (*n* = 50)	Survival (*n* = 42)	Expired (*n* = 8)	*p* Value
Male	18 (36%)	15 (35.71%)	3 (37.50%)	1.000
Age	64.33 ± 11.27	62.52 ± 10.91	73.82 ± 8.34	0.008 **
Comorbidities				
Cardiovascular disease	30 (60%)	24 (57.14%)	6 (75.00%)	0.450
DM	34 (68%)	28 (66.67%)	6 (75.00%)	1.000
Hyperlipidemia	10 (20%)	8 (19.05%)	2 (25.00%)	0.653
Gout	6 (12%)	5 (11.90%)	1 (12.50%)	1.000
CVA	1 (2%)	1 (2.38%)	0(0%)	1.000
COPD	2 (4%)	2 (4.76%)	0(0%)	1.000
GI disease	18 (36%)	12 (28.57%)	6 (75.00%)	0.019 *
Chronic renal failure	20 (40%)	16 (38.10%)	4 (50.00%)	0.697
PAOD	1 (2%)	1 (2.38%)	0(0%)	1.000
Transplant	1 (2%)	1 (2.38%)	0(0%)	1.000
GU disease	15 (30%)	15 (35.71%)	0(0%)	0.086
Immune disorder	8 (16%)	7 (16.67%)	1 (12.50%)	1.000
Tumor	7 (14%)	7 (16.67%)	0(0%)	0.580
Risk factors				
DM	34 (68%)	28 (66.67%)	6 (75.00%)	1.000
Obstructive uropathy	16 (32%)	14 (33.33%)	2 (25.00%)	1.000
Symptoms				
Fever	25 (50%)	22 (52.38%)	3 (37.50%)	0.702
Flank pain	20 (40%)	18 (42.86%)	2 (25.00%)	0.450
Abdominal pain	10 (20%)	8 (19.05%)	2 (25.00%)	0.653
Conscious change	10 (20%)	8 (19.05%)	2 (25.00%)	0.653
GI symptoms	15 (30%)	13 (30.95%)	2 (25.00%)	1.000
LUTS	5 (10%)	5 (11.90%)	0(0%)	0.577
Respiratory	4 (8%)	3 (7.14%)	1 (12.50%)	0.514
Nonspecific	5 (10%)	3 (7.14%)	2 (25.00%)	0.176
Vital signs				
SBP	131.72 ± 32.25	130.62 ± 32.11	137.50 ± 34.59	0.608
DBP	79.26 ± 24.55	79.21 ± 21.43	79.50 ± 39.09	0.790
MAP	96.75 ± 25.34	96.35 ± 23.60	98.83 ± 35.00	0.933
HR	101.94 ± 20.93	100.71 ± 21.18	108.38 ± 19.57	0.486
RR	19.36 ± 2.89	19.05 ± 2.33	21.00 ± 4.81	0.433
BT	36.88 ± 1.13	36.94 ± 1.00	36.61 ± 1.74	0.730
GCS	14.06 ± 2.32	14.31 ± 1.89	12.75 ± 3.77	0.227
SpO_2_	98.32 ± 4.04	98.95 ± 1.79	95.00 ± 8.96	0.646

* *p* < 0.05, ** *p* < 0.01, Statistically significant. Abbreviations: BT, Body temperature; COPD, Chronic obstructive pulmonary disease; CVA, Cerebrovascular accident; DBP, Diastolic blood pressure; DM, Diabetes mellitus; GCS, Glasgow coma scale; GI: Gastrointestinal; GU, Genitourinary; HR, Heart rate; LUTS, Lower urinary tract symptoms; MAP, Mean blood pressure; RR, Respiratory rate; SBP, Systolic blood pressure.

**Table 2 jcm-11-07299-t002:** Laboratory data of EPN.

Laboratory Data	All (*n* = 50)	Survival (*n* = 42)	Expired (*n* = 8)	*p* Value
Biochemistry				
Albumin (g/dL)	2.52 ± 0.61	2.58 ± 0.58	2.21 ± 0.67	0.075
Total bilirubin (mg/dL)	0.96 ± 0.94	0.93 ± 0.97	1.11 ± 0.83	0.432
ALK-P (U/L)	201.34 ± 134.26	202.39 ± 141.20	196.63 ± 104.96	0.737
AST (U/L)	47.43 ± 32.78	45.26 ± 33.56	57.75 ± 28.46	0.143
ALT (U/L)	30.86 ± 26.63	29.48 ± 26.48	38.13 ± 28.04	0.275
LDH (U/L)	282.58 ± 127.80	265.61 ± 112.40	377.60 ± 179.10	0.142
BUN (mg/dL)	41.00 ± 34.30	39.00 ± 36.37	51.50 ± 18.42	0.027 *
Cr (mg/dL)	2.39 ± 1.82	2.35 ± 1.95	2.57 ± 0.97	0.236
Na (mEq/L)	132.16 ± 6.88	132.67 ± 6.74	129.50 ± 7.45	0.527
K (mEq/L)	4.67 ± 4.89	3.98 ± 0.94	8.26 ± 12.03	0.371
Ca (mg/dL)	8.25 ± 1.05	8.29 ± 1.12	8.05 ± 0.66	0.345
CRP (mg/dL)	20.14 ± 12.97	20.02 ± 13.31	20.81 ± 11.83	0.729
Lactate (mg/dL)	26.96 ± 26.01	21.26 ± 20.78	50.44 ± 33.32	0.020 *
Glucose (mg/dL)	212.56 ± 114.97	211.67 ± 99.07	217.25 ± 187.12	0.671
Blood cell counts				
WBC (×10^3^ counts/mm^3^)	16.77 ± 9.76	16.48 ± 9.35	18.27 ± 12.28	0.750
Hemoglobin	10.55 ± 1.74	10.44 ± 1.79	11.09 ± 1.47	0.299
Platelet (×10^3^ counts/mm^3^)	292.04 ± 183.12	306.17 ± 183.61	217.88 ± 172.59	0.258
Band (%)	2.72 ± 5.25	2.40 ± 5.20	4.38 ± 5.55	0.339
Neutrophil (Segment) (%)	91.88 ± 27.64	93.03 ± 29.64	84.97 ± 6.58	0.649
PT (s)	11.72 ± 2.03	11.56 ± 1.87	12.54 ± 2.68	0.350
APTT (s)	33.71 ± 10.95	33.57 ± 10.97	34.53 ± 11.70	0.934
Arterial blood gas				
pH	7.39 ± 0.10	7.39 ± 0.10	7.37 ± 0.09	0.516
PaCO_2_ (mmHg)	35.13 ± 7.50	35.52 ± 7.77	33.31 ± 6.25	0.452
PaO_2_ (mmHg)	75.86 ± 48.70	77.49 ± 51.20	68.34 ± 36.66	0.805
HCO_3_^−^ (mmol)	20.92 ± 6.15	21.35 ± 6.50	18.94 ± 3.90	0.249

* *p* < 0.05, Statistically significant. Abbreviations: ALK-P, Alkaline phosphatase; ALT, Alanine aminotransferase; AST, Aspartate aminotransferase; BUN, Blood urea nitrogen; CRP, C-reactive protein; Ca, Calcium; Cr, Creatinine; K, Potassium; LDH, Lactate dehydrogenase; Na, Sodium; PT, Prothrombin time; APTT, Activated partial prothrombin time.

**Table 3 jcm-11-07299-t003:** Scoring systems for predicting the clinical outcomes of EPN.

Scoring Systems	All (*n* = 50)	Survival (*n* = 42)	Expired (*n* = 8)	*p* Value
GAPS	20.30 ± 2.94	20.74 ± 2.63	18.00 ± 3.55	0.023 *
MEDS	6.18 ± 5.29	4.64 ± 3.67	14.25 ± 5.34	<0.001 **
MEWS	3.14 ± 1.97	2.93 ± 1.87	4.25 ± 2.25	0.087
NEWS	4.20 ± 3.09	3.64 ± 2.33	7.13 ± 4.85	0.046 *
RAPS	2.22 ± 2.12	2.00 ± 1.83	3.38 ± 3.16	0.252
REMS	6.34 ± 2.62	5.81 ± 1.97	9.13 ± 3.87	0.024 *
qSOFA	0.46 ± 0.65	0.40 ± 0.59	0.75 ± 0.89	0.272
WPS	2.14 ± 2.19	1.74 ± 1.42	4.25 ± 3.96	0.088

* *p* < 0.05, ** *p* < 0.01, Statistically significant. Abbreviations: GAPS, Glasgow admission prediction score; MEDS, Mortality in Emergency Department Sepsis; MEWS, Modified Early Warning Score; NEWS, National Early Warning Score; RAPS, Rapid Acute Physiology Score; REMS, Rapid Emergency Medicine Score; qSOFA, quick Sepsis-related Organ Failure Assessment; WPS, Worthing Physiological Scoring system.

**Table 4 jcm-11-07299-t004:** Microbiology of EPN.

Bacteria/Culture	Urine (*n* = 48)	Blood (*n* = 45)	Pus/Tissue (*n* = 32)
*Escherichia coli*	20	13	14
*Klebsiella pneumoniae*	4	9	11
*Staphylococcus* *aureus*	1	0	1
*Enterobacter* *cloacae*	1	1	1
*Pseudomonas* *aeruginosa*	2	1	0
*Streptococcus pneumoniae*	2	3	1
*Enterococcus* *faecalis*	2	0	1
*Candida* *albicans*	3	0	0

Cultures of urine, blood, pus, and/-or tissue were collected in 48, 45, and 32 patients, respectively. The positive urine culture is defined as the number > 10^5^ CFU/mL.

**Table 5 jcm-11-07299-t005:** Antibiotics, management, and clinical course of EPN.

Patients	All (*n* = 50)	Survival (*n* = 42)	Expired (*n* = 8)	*p* Value
Antibiotics Category				
Cephalosporin ^f^	36 (72%)	31 (73.81%)	5 (62.50%)	0.670
Beta-lactam ^f^	13 (26%)	12 (28.57%)	1 (12.50%)	0.662
Sulfa drug ^f^	1 (2%)	1 (2.38%)	0 (0%)	1.000
Metronidazole ^f^	6 (12%)	5 (11.90%)	1 (12.50%)	1.000
Penicillin ^f^	4 (8%)	2 (4.76%)	2 (25.00%)	0.115
Quinolone ^f^	5 (10%)	4 (9.52%)	1 (12.50%)	1.000
Vancomycin ^f^	2 (4%)	1 (2.38%)	1 (12.50%)	0.297
Management				0.062
Antibiotics only	11 (22%)	8 (19.05%)	3 (37.50%)	0.351
Antibiotics + PCD	25 (50%)	23 (54.76%)	2 (25.00%)	0.247
Antibiotics + Surgery	7 (14%)	7 (16.67%)	0(0%)	0.580
Antibiotics + PCD + Surgery	7 (14%)	4 (9.52%)	3 (37.50%)	0.071
Clinical course				
SIRS/Shock ^f^	25 (50%)	17 (40.48%)	8 (100%)	0.004 **
Respiratory failure ^f^	17 (34%)	11 (26.19%)	6 (75.00%)	0.013 *
ICU admission ^f^	17 (34%)	11 (26.19%)	6 (75.00%)	0.013 *
Length of hospital stay	25.58 ± 16.82	24.33 ± 14.04	32.13 ± 27.71	0.562

Chi-Square test. ^f^ Fisher’s Exact test. Mann–Whitney U-test.* *p* < 0.05, ** *p* < 0.01. Continuous data were expressed as mean ± SD. Categorical data were expressed as number and percentage. Abbreviations: PCD, Percutaneous drainage.

**Table 6 jcm-11-07299-t006:** Hazard ratios and 95% confidence interval of univariate analyses.

Characteristics	Hazard Ratios	95% Confidence Interval	*p* Value
Age (years)	1.107	1.016–1.207	0.020 *
Male	1.213	0.271–5.434	0.801
Clinical conditions			
Shock	50.643	0.102–25,238.144	0.216
Respiratory failure	2.804	0.529–14.870	0.226
Vital signs			
Systolic blood pressure (mmHg)	1.009	0.985–1.034	0.452
Mean arterial pressure (mmHg)	1.008	0.979–1.037	0.585
Heart rate (bpm)	1.027	0.989–1.067	0.165
Respiratory rate (bpm)	1.230	1.034–1.463	0.012 *
Body temperature (°C)	1.056	0.581–1.920	0.857
Glasgow coma scale	0.806	0.655–0.991	0.041 *
SpO_2_ (%)	0.802	0.704–0.915	0.001 **
Comorbidities			
Cardiovascular disease	1.472	0.285–7.615	0.645
Diabetes mellitus	1.057	0.204–5.475	0.947
Chronic kidney disease	2.631	0.580–11.931	5.475
Immune disorder	0.806	0.097–6.711	0.842
Tumor	0.035	0.000–141.610	0.430
Laboratory data			
White blood cell (counts/uL)	1.000	1.000–1.000	0.777
Hemoglobin (g/dL)	1.430	0.838–2.439	0.189
Platelet (×10^3^ counts/uL)	1.000	1.000–1.000	0.097
Albumin (g/dL)	0.355	0.073–1.736	0.201
ALK-P (U/L)	0.997	0.989–1.004	0.385
AST (U/L)	1.021	0.998–1.044	0.077
ALT (U/L)	1.009	0.985–1.035	0.454
Lactate dehydrogenase (U/L)	1.007	1.000–1.014	0.038 *
C-reactive protein (mg/dL)	1.005	0.947–1.066	0.876
Lactate (mg/dL)	1.031	1.009–1.053	0.005 **
PT (sec)	1.323	1.002–1.748	0.049 *
APTT (sec)	1.008	0.956–1.063	0.770
Total bilirubin (mg/dL)	1.186	0.632–2.227	0.595
pH	1.914	0.002–1675.545	0.851
HCO_3_ (mmol/L)	0.942	0.810–1.094	0.433
Scoring systems			
GAPS	0.784	0.643–0.957	0.017 *
MEDS	1.833	1.216–2.764	0.004 **
MEWS	1.223	0.959–1.558	0.104
NEWS	1.632	1.218–2.188	0.001 **
RAPS	1.465	1.026–2.092	0.036 *
REMS	1.913	1.303–2.811	0.001 **
qSOFA	3.577	1.146–11.162	0.028 *
WPS	1.710	1.246–2.348	0.001 **
Symptoms			
Fever	0.713	0.159–3.186	0.657
Flank pain	0.216	0.026–1.805	0.157
Abdominal pain	1.532	0.295–7.952	0.612
Conscious change	1.427	0.276–7.377	0.671
GI symptoms	0.928	0.180–4.798	0.929
LUTS	0.042	0.000–2573.964	0.574
Respiratory	2.164	0.259–18.055	0.476
Nonspecific	0.980	0.117–8.206	0.985

* *p* < 0.05, ** *p* < 0.01, Statistically significant. Abbreviations: ALK-P, alkaline phosphatase; ALT, Alanine aminotransferase; AST, Aspartate aminotransferase; GI, Gastrointestinal; LUTS, Lower urinary tract symptoms; PT, Prothrombin time; APTT, Activated partial prothrombin time. GAPS, Glasgow Admission Prediction Score; MEDS, Mortality in Emergency Department Sepsis; MEWS, Modified Early Warning Score; NEWS, National Early Warning Score; RAPS, Rapid Acute Physiology Score; REMS, Rapid Emergency Medicine Score; qSOFA, quick Sepsis-related Organ Failure Assessment; WPS, Worthing Physiological Scoring system.

**Table 7 jcm-11-07299-t007:** Hazard ratios and 95% confidence interval of univariate analysis and multivariate Cox regression analysis.

	Univariate			Multivariate ^a^		
Variables	HR	95% CI	*p* Value	HR	95% CI	*p* Value
GCS	0.806	0.655–0.991	0.041 *	0.834	0.671–1.036	0.101
SpO_2_	0.802	0.704–0.915	0.001 **	0.782	0.676–0.905	0.001 **
GAPS	0.784	0.643–0.957	0.017 *	0.807	0.656–0.993	0.043 *
MEDS	1.833	1.216–2.764	0.004 **	1.850	1.207–2.836	0.005 **
NEWS	1.632	1.218–2.188	0.001 **	1.638	1.209–2.220	0.001 **
RAPS	1.465	1.026–2.092	0.036 *	1.404	0.986–1.999	0.060
REMS	1.913	1.303–2.811	0.001 **	1.404	0.986–1.999	0.060
qSOFA	3.577	1.146–11.162	0.028 *	3.275	1.038–10.336	0.043 *
WPS	1.710	1.246–2.348	0.001 **	1.699	1.226–2.356	0.001 **
Age	1.107	1.016–1.207	0.020 *	1.101	1.010–1.200	0.029 *
Pus/Tissue	0.115	0.022–0.605	0.011 *	0.098	0.018–0.521	0.006
RR	1.230	1.034–1.463	0.012 *	1.198	0.995–1.441	0.057
LDH	1.007	1.000–1.014	0.038 *	1.008	1.000–1.016	0.046 **
Lactate	1.031	1.009–1.053	0.005 **	1.030	1.007–1.053	0.010 *
PT	1.323	1.002–1.748	0.049 *	1.421	1.050–1.922	0.023

* *p* < 0.05 and ** *p* < 0.01, Statistically significant. ^a^ Multivariate Cox regression analysis adjusted by admission to ICU; Abbreviations: GCS, Glasgow coma scale; LDH, Lactate dehydrogenase; RR, Respiratory rate; PT, Prothrombin time. GAPS, Glasgow Admission Prediction Score; MEDS, Mortality in Emergency Department Sepsis; MEWS, Modified Early Warning Score; NEWS, National Early Warning Score; RAPS, Rapid Acute Physiology Score; REMS, Rapid Emergency Medicine Score; qSOFA, quick Sepsis-related Organ Failure Assessment; WPS, Worthing Physiological Scoring system.

**Table 8 jcm-11-07299-t008:** The AUC of ROC, cut-off point, sensitivity specificity, positive predictive value (PPV), negative predictive value (NPV), accuracy, and standard error (SE) of MEDS, NEWS, REMS, and GAPS to predict the mortality risk.

Scores	AUC	Cut-Off Point	Sensitivity	Specificity	PPV	NPV	Accuracy	SE	*p* Value
MEDS	0.932	12	75.0%	95.2%	75.0%	95.2%	92.0%	0.047	<0.001 **
REMS	0.747	10	62.5%	100.0%	100.0%	93.3%	94.0%	0.124	0.028 *
NEWS	0.720	8	50.0%	97.6%	80.0%	91.1%	90.0%	0.122	0.050 *
GAPS	0.747	22	100%	40.5%	24.2%	100%	50%	0.084	0.028 *

* *p* < 0.05 and ** *p* < 0.01, Statistically significant; Abbreviations: AUC, Area under the curve; GAPS, Glasgow admission prediction score; MEDS, Mortality in Emergency Department Sepsis; NEWS, National Early Warning Score; NPV, negative predictive value; PPV, positive predictive value; REMS, Rapid Emergency Medicine Score; ROC, Receiver operating characteristic curve; SE, Standard error.

## Data Availability

Readers can access the data and material supporting the study’s conclusions by contacting S.-Y.H. at song9168@pie.com.tw.

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
