# Peer review of "Performance of Scoring Systems in Predicting Clinical Outcomes of Patients with Emphysematous Pyelonephritis: A 14-Year Hospital-Based Study"

_jcm, 2022, doi:10.3390/jcm11247299_

Round 1

Reviewer 1 Report

Chen 2022 subm JCM-Review..scoring systems in pred clinical outcomes of pat with emphys PN-A 14-year hospital-based study

This is a very interesting, retrospective study  predicting outcomes in patients with EPN.The authors analysed 50 patients and used several scoring systems. Three of the scoring systems were good predictors for mprtality risk of EPN.

The study is well designed, performed, and presented. Since EPN is a fairly rare infection prospective studies are practically not feasible. Howver, the results of this retrospüective study are very interesting.I only have a few minor comments/questions:

1. Line 79: The current meta-analysis reported mortality rates up to 20-40% (Reference is missing)

2. Line 85: CO2, not CO2.

3. Line 90: …by Hudson et al (Reference is missing)

4. Line 172: The positive rate of urine culture was 37 (74%) of 48 patients. The reader would like to know, what number of CFU/ml means positive in this study, e.g. >103 CFU/ml, or >104 CFU/ml, or >105 CFU/ml.

5. Table 4. Urine (n=48) all mentioned bacterial species are together 35, however the authors stated  that „The positive rate of urine culture was 37 (74%) of 48 patients“. Which means 2 are missing; please check it.

I have no further comments

Author Response

According to the minor sugestions and comments of Reviewer, the authors reply as below:

1. Line 79: The current meta-analysis reported mortality rates up to 20-40%. (Reference is missing)

Reply:

Thanks for reviewer’s comments. The authors make changes in the manuscript.

The authors add “Desai, R.; Batura, D. A systematic review and meta-analysis of risk factors and treatment choices in emphysematous pyelonephritis. Int Urol Nephrol 2022, 54, 717-736” as reference [4].

2. Line 85: CO2, not CO2.

Reply:

Thanks for reviewer’s comments. The authors make changes in the manuscript.

The authors revise “Enteric gram-negative facultative anaerobes were common microorganisms of EPN, fermented glucose within the urine to produce gases, including nitrogen, hydrogen, and carbon dioxide (CO2), resulting in emphysematous necrosis.” in the manuscript. 

3. Line 90: …by Hudson et al (Reference is missing)

Reply:

Thanks for reviewer’s comments. The authors make changes in the manuscript.

The authors add “Hudson, M.A.; Weyman, P.J.; van der Vliet, A.H.; Catalona, W.J. Emphysematous pyelonephritis: successful management by percutaneous drainage. J Urol 1986, 136, 884-886” as reference [9].

4. Line 172: The positive rate of urine culture was 35 (72.9%) of 48 patients. The reader would like to know, what number of CFU/ml means positive in this study, e.g. >103 CFU/ml, or >104 CFU/ml, or >105 CFU/ml.

Reply:

Thanks for reviewer’s comments. The authors make changes in Table 4.

The authors add “The positive urine culture is defined the number >105 CFU/ml.” in below the Table 4. 

5. Table 4. Urine (n=48) all mentioned bacterial species are together 35, however the authors stated that “The positive rate of urine culture was 37 (74%) of 48 patients“. Which means 2 are missing; please check it.

Reply:

Thanks for reviewer’s comments. The authors make changes in the section of Microbiology in the manuscript.

The authors revise the description “The positive rate of blood culture was 27 (60.0%) of 45 patients. The positive rate of urine culture was 35 (72.9%) of 48 patients. The positive rate of pus or tissue was 29 (90.6%) of 32 patients” in the manuscript.

Thanks Reviewer's suggestions and comments to strengthen our manuscript. 

Sung-Yuan Hu,

Department of Emergency Medicine, Taichung Veterans General Hospital, Taiwan.

Reviewer 2 Report

Introduction paragraph: "The current meta-analysis reported mortality 78 rates up to 20-40%. Radiological investigations ensured the diagnosis of EPN, including 79 abdominal ultrasound (US) and computed tomography (CT)".

You cannot refer to your study before the presentation of its results. Either erase this sentence or remove it to Discussion paragraph.

Author Response

According to the minor suggestions and comments of Reviewer, the authors reply as below:

Reply:

Thanks for reviewer’s comments. The authors make changes in the manuscript.

1. The authors add “Desai, R.; Batura, D. A systematic review and meta-analysis of risk factors and treatment choices in emphysematous pyelonephritis. Int Urol Nephrol 2022, 54, 717-736.” as reference [4].

2. The authors add “Tamburrini, S.; Lugarà, M.; Iannuzzi, M.; Cesaro, E.; Simone, F.D.; Biondo, D.D.; Toto, R.; Iulia, D.; Marrone, V.; Faella, P.; et al. Pyonephrosis Ultrasound and Computed Tomography Features: A Pictorial Review. Diagnostics (Basel) 2021, 11, 331.” as reference [5].

Thanks Reviewer's suggestions and comments to strengthen our manuscript. 

Sung-Yuan Hu,

Department of Emergency Medicine, Taichung Veterans General Hospital, Taiwan.
